# Temperature-Dependent Feedback Operations of Triple-Gate Field-Effect Transistors

**DOI:** 10.3390/nano14060493

**Published:** 2024-03-09

**Authors:** Taeho Park, Kyoungah Cho, Sangsig Kim

**Affiliations:** Department of Electrical Engineering, Korea University, 145 Anam-ro, Seongbuk-gu, Seoul 02841, Republic of Korea; pth0420@korea.ac.kr

**Keywords:** feedback field-effect transistor (FBFET), latch-up phenomenon, positive feedback loop, temperature-dependent, TCAD simulation

## Abstract

In this study, we examine the electrical characteristics of triple-gate feedback field-effect transistors (TG FBFETs) over a temperature range of −200 °C to 280 °C. With increasing temperature from 25 °C to 280 °C, the thermally generated charge carriers increase in the channel regions such that a positive feedback loop forms rapidly. Thus, the latch-up voltage shifts from −1.01 V (1.34 V) to −11.01 V (10.45 V) in the *n*-channel (*p*-channel) mode. In contrast, with decreasing temperature from 25 °C to −200 °C, the thermally generated charge carriers decrease, causing a shift in the latch-up voltage in the opposite direction to that of the increasing temperature case. Despite the shift in the latch-up voltage, the TG FBFETs exhibit ideal switching characteristics, with subthreshold swings of 6.6 mV/dec and 7.2 mV/dec for the *n*-channel and *p*-channel modes, respectively. Moreover, the memory window widens with increasing temperature. Specifically, at temperatures above 85 °C, the memory windows are wider than 3.05 V and 1.42 V for the *n*-channel and *p*-channel modes, respectively.

## 1. Introduction

With the advent of the big data era, in-memory computing (IMC) has emerged as a promising computing paradigm for data-intensive applications such as artificial intelligence and Internet of Things [1,2,3]. IMC presents a solution to the memory wall of the traditional von Neumann architecture by merging the functions of logic and memory units, which is a novel solution to reduce the latency and energy consumption associated with data movement [4,5,6,7]. Among the various emerging devices [8,9,10,11,12], feedback field-effect transistors (FBFETs) have attracted attention as suitable devices for IMC implementation, because they perform hybrid switching and memory storage [13,14,15]. Moreover, multiple-gate FBFETs provide enhanced flexibility in circuit design owing to their reconfigurable operation between the *p*- channel and *n*-channel modes in single devices, which is advantageous for reducing the number of devices required for complex IMC circuit designs [16]. To extend the application of versatile multiple-gate FBFETs, it is essential to investigate the effect of temperature on the operating characteristics of FBFETs, because temperature is one of the primary factors affecting the reliability of electronic devices. On the other hand, the effect of temperature has also been investigated on the electrical characteristics of 2D-material-based FETs, which have attracted considerable attention as components for high-scalable and low-power electronic devices [17,18]. Therefore, in this study, we investigate the electrical characteristics of triple-gate FBFETs (TG FBFETs) at temperatures ranging from −200 °C to 280 °C through technology computer-aided design (TCAD) simulations.

## 2. Simulation Methods

Figure 1 shows a schematic of a TG FBFET with a *p*^+^-*i*-*n*^+^ nanowire structure and an intrinsic channel region surrounded by two program gates (PGs) and one control gate (CG). The two PGs are electrically connected to each other and control the channel mode through electrostatic doping induced by voltage polarity, and the CG functions as a switching gate. In the TG FBFET design, the lengths of the drain (*L*_D_), source (*L*_S_), and gate (*L*_G_) regions were set to 50 nm. The channel (*T*_Si_) and gate oxide (*T*_OX_) thicknesses were 10 nm and 2 nm, respectively. The three gates were separated by a 10 nm gap (*L*_GAP_), and heavily doped polysilicon was used as the gate material. The doping concentrations of the *p*-type drain and *n*-type source regions were heavily doped at 1 × 10^20^ cm^−3^, while the intrinsic channel region was lightly *p*-type doped at 1 × 10^15^ cm^−3^. Our simulations were performed with 2D structure using the commercial device simulator Synopsys Sentaurus (Version T-2022.03). The physical model included Fermi–Dirac statistics, high-field-saturation mobility, inversion and accumulation layer mobility, and Slotboom bandgap narrowing. A thermodynamic model was applied to account for the temperature dependence of the carrier transport mechanism. Moreover, we considered Shockley–Read–Hall (SRH) recombination using a temperature-dependent model and Auger recombination.

## 3. Results and Discussion

Figure 2 shows the energy band diagrams of the TG FBFET with a *p*^+^-*i*-*n*^+^ structure. The channel mode is determined by the voltage polarity applied to the two PGs; a positive gate bias induces electrons (*n**), whereas a negative gate bias induces holes (*p**) in the channel region. When *V*_PG_ = −3 V and *V*_CG_ = 2 V, the channel region is doped electrostatically to form a *p**-*n**-*p** structure and operates in the *p*-channel mode. The energy level of the *p** region near the drain is reduced due to hole injection from the drain, which is responsible for the difference between the energy levels in the two *p** regions, even with the same *V*_PG_ of −3 V. Initially, the potential barrier created by the electrostatic doping of *V*_PG_ and *V*_CG_ blocks the injection of charge carriers from the drain/source regions into the channel region. However, as *V*_CG_ sweeps from 2 V to 0 V with a drain-to-source voltage (*V*_DS_) of 1.2 V, holes are injected from the drain into the channel regions through the lowered potential barrier. The accumulation of holes in the *p** region near the source lowers the potential barrier for electrons, allowing electrons to flow from the source to the channel regions. These electrons accumulated in the potential well of the *n** region further lower the potential barrier for holes and accelerate hole injection into the channel region. The continuous injection and accumulation of charge carriers lead to the collapse of the potential barriers, activating the positive feedback (PF) loop and switching the device to the on state with the latch-up phenomenon. Here, the *V*_CG_ at which the latch-up phenomenon occurs is defined as the latch-up voltage. For the *n*-channel mode, an electrostatically induced *n**-*p**-*n** doping structure is formed in the channel region at *V*_PG_ = 3 V and *V*_CG_ = −2 V. The difference in the energy levels of the two *n** regions with the same *V*_PG_ is due to the injection of electrons from the source region, which increases the energy level in the *n** region adjacent to the source region. While *V*_CG_ sweeps forward from −2 V to 0 V at a source-to-drain voltage (*V*_SD_) of −1.2 V, the continuous interaction between charge carriers and potential barrier induces a PF loop, leading the device to switch to the on state through the latch-up phenomenon. Once the PF loop is activated, the on state is maintained until the charge carriers in the channel region are removed. To turn off the device via the latch-down phenomenon, it is necessary to eliminate the PF loop, which is achieved by applying a sufficient *V*_CG_, defined as the latch-down voltage required to release charge carriers inside the channel region. The TG FBFET maintains the on state before eliminating the PF loop such that it has a memory window in the transfer curve.

The transfer characteristics of the TG FBFET in a temperature range from −200 °C to 25 °C are shown in Figure 3a. A *V*_SD_ value of −1.2 V (*V*_DS_ = 1.2 V) and a *V*_PG_ value of 3.0 V (−3.0 V) were applied to operate the device in the *n*-channel (*p*-channel) mode. While sweeping *V*_CG_ from −3.0 V (3.0 V) to 0.0 V in the *n*-channel (*p*-channel) mode, an abrupt increase in the drain current occurs at *V*_CG_, the latch-up voltage. Steep switchings are present at temperatures of −200 °C to 25 °C in both the *n*-channel and *p*-channel modes, as shown in Figure 3a. The values of subthreshold swings (*SS*s), defined as the minimum values of ∂*V*_CG_/∂log(*I*_DS_) in the *n*-channel (*p*-channel) mode, are 1.9, 2.0, 2.1, 2.9, and 6.6 mV/dec (2.1, 2.4, 4.2, 7.2, and 6.3 mV/dec) at temperatures of 25, 0, −25, −100, and −200 °C, respectively. Although the *SS* values decrease with decreasing temperature, they are still lower than those of other steep-switching devices [19,20,21]. As the temperature decreases, the latch-up voltage shifts toward 0 V in both the *n*-channel and *p*-channel modes, indicating that the PF loop was formed later. The formation of the PF loop depends not only on the number of charge carriers injected into the channel by *V*_CG_, but also on the number of accumulated charge carriers within the potential well through diffusion. At low temperatures, the thermal energy is insufficient to fully activate acceptors in the *p*-doped drain region and donors in the *n*-doped source region, which is particularly pronounced in heavily doped Si [22,23]. Considering a heavily doped source/drain of 1 × 10^20^ cm^−3^, the ionization rate of the dopants in the source and drain regions decreases significantly as the temperature decreases, and as a result, the amount of charge carriers injected into the channel also decreases significantly. Moreover, a decrease in thermal velocity at low temperatures leads to a decrease in the diffusion current, thereby reducing the accumulation of charge carriers in the potential well. Thus, as the temperature decreases, *V*_CG_ shifts more positively in the *n*-channel mode (−1.01 → −0.96 → −0.90 → −0.74 → −0.58 V) and negatively in the *p*-channel mode (1.34 → 1.27 → 1.21 → 1.05 → 0.91) to obtain enough charge carriers to generate a PF loop.

Figure 3b shows the transfer characteristics of the TG FBFET at temperatures ranging from 25 °C to 280 °C. Here, the sweep of |*V*_CG_| starts at 15 V, because the TG FBFET at temperatures above 200 °C attains the normally on state under the condition of the |*V*_CG_| at low temperatures. Nevertheless, the TG FBFET is in the normally on state at 280 °C in the *n*-channel mode. By contrast, the *SS* values are maintained in a range of 1.4~3.3 mV/dec even at high temperatures. According to a previous study [24], charge carriers with high kinetic energies can surpass the potential barriers formed in the channel region at high temperatures. This leads to the injection of charge carriers from the source or drain regions into the channel region, thereby activating a thermally induced PF loop. To maintain the off state, a large voltage |*V*_CG_| is required to increase the initial height of the potential barrier that blocks the charge carriers. In contrast to the operation of the TG FBFET at low temperatures, a PF loop occurs more rapidly owing to the thermal stimulation and generation of charge carriers at high temperatures. The latch-up voltage in the *n*-channel mode shifts from −1.01 V to −11.01 V as the temperature increases from 25 °C to 240 °C. By contrast, the latch-up voltage in the *p*-channel mode shifts from 1.34 V to 10.45 V as the temperature increases from 25 °C to 280 °C. Despite the high potential barrier formed by the |*V*_CG_| of 15 V, thermally stimulated charge carriers are injected in the drain region for the *n*-channel mode and in the source region for the *p*-channel mode, and then, accumulate in the potential well for holes (electrons), resulting in a reduction in the potential barrier for electrons (holes) under CG. Additionally, thermally generated electrons and holes further reduce the potential barrier, promoting the PF loop; the thermally generated electron–hole mechanism will be discussed later. Figure 3c shows the latch-up/down voltage shifts that occurred positively (negatively) in the *p*-channel (*n*-channel) modes. The memory window, which is defined as the difference between the latch-up and latch-down voltages, widens at high temperatures, as shown in Figure 3d. As the temperature increases from −200 °C to 280 °C, the concentration of charge carriers in the channel region increases exponentially, which contributes to maintaining a PF loop such that a larger |*V*_CG_| is required to eliminate this loop. Thus, the latch-down voltage shifts in the more negative (positive) direction in the *n*-channel (*p*-channel) mode; consequently, a higher temperature creates a wider memory window. At a temperature of 85 °C, the memory windows are 3.05 V and 1.42 V for the *n*-channel and *p*-channel modes, respectively. By contrast, open memory windows become apparent in the *n*-channel mode at temperatures above 200 °C and in the *p*-channel mode at a temperature of 280 °C.

Figure 4 shows the thermal generation mechanism of the TG FBFET. In the *p*-channel mode, the potential barriers and wells are induced in the energy band in the form of a *p*^+^-*p**-*n**-*p**-*n*^+^ structure that prevents the flow of electrons and holes, as shown in Figure 4a. The built-in potential voltage (*V*_bi_) of the isotype junction (J_1_) is formed between the drain and *p** regions. The *p** region electrostatically doped with negative *V*_PG_ is expressed by (1):(1)$$V_{\mathrm{bi}}=\frac{kT}{q}\ln\frac{N_{A}}{N_{A}^{*}}$$
where $N_{A}$ is the acceptor concentration in the drain region and $N_{A}^{*}$ is the electrostatic doping concentration in the *p** region. Owing to the logarithmic dependence of *V*_bi_ on the doping concentration, the potential barrier between the drain and *p** regions has a negligible effect on the flow of holes from the drain to the *p** regions. Accordingly, the drain extends to the *p*^+^-*p** region in *p*-channel mode. When a positive voltage is applied to *V*_DS_, a reverse-biased junction (J_3_) is induced between the *n** region under CG and the *p** region under PG adjacent to the source region. In a classical *p*-*n* junction diode, the reverse leakage current, which is the sum of the saturation current and thermal generation current, is governed by diffusion and carrier generation mechanisms [25]. In this case, the diffusion component, referred to as the saturation current (*I*_S_), is given by (2):(2)$$I_{S}=qAn_{i}^{2}\left(\frac{D_{p}}{W_{n^{*}}N_{D}}+\frac{D_{n}}{W_{p^{*}}N_{A}}\right)$$
where *A* is the junction area, $n_{i}$ is intrinsic carrier concentration, *D*_n_ is the electron diffusion coefficient, *D*_p_ is the hole diffusion coefficient, *W*_n*_ is the width of the *n** region, and *W*_p*_ is the width of the *p** region. The intrinsic carrier concentration and diffusion coefficients are significantly influenced by temperature, indicating that the saturation current depends on the temperature change. Another component of the reverse leakage current is the thermal generation of electrons and holes through defects in the depletion region [26]. The thermal generation current (*I*_gen_) in the depletion region can be expressed as
(3)$$g_{\mathrm{th}}≅\frac{n_{i}}{2\tau_{0}}$$
(4)$$I_{\mathrm{gen}}=qAg_{\mathrm{th}}W_{d}$$
where *g*_th_ is the thermal generation rate, *τ*_0_ is the minority carrier lifetime, and *W*_d_ is the depletion width. At high temperatures, the high intrinsic carrier concentration and short minority carrier lifetime rapidly increase the thermal generation current [26]. Unlike a classical *p*-*n* diode, in the *p*-channel mode, electrons in the reverse leakage current are blocked by the potential barrier formed in J_2_ and accumulate in the potential well formed in the *n** region, whereas holes are blocked by the potential barrier formed in J_4_ and accumulate in the potential well formed in the *p** region. The accumulated charge carriers trigger perturbations in the energy level in both the *n** and *p** regions close to the source region, such that the PF loop occurs more rapidly. In the *n*-channel mode, the potential barriers and wells are built into the energy band in the form of a *p*^+^-*n**-*p**-*n**-*n*^+^ structure. With a mechanism identical to that of the *p*-channel mode, the source region extends to the *n**-*n*^+^ region in the *n*-channel mode. The negative voltage applied to *V*_SD_ induces a reverse-biased junction (J_6_) between the *p** region under the CG and *n** region under the PG adjacent to the drain. The electrons and holes generated in J_6_ accumulate in the potential wells formed in the *n** and *p** regions, respectively, causing perturbations in the corresponding energy levels.

Contour maps of the electron–hole recombination and generation of the TG FBFET at various temperatures are shown in Figure 5a, where the negative and positive signs of the SRH recombination rate indicate the generation and recombination rates, respectively. The simulation results agree well with the thermal generation mechanism shown in Figure 4. The contour map of the *n*-channel mode at 280 °C represents high SRH recombination, indicating that the potential barrier collapses owing to the thermally induced PF loop. The electron–hole generation rates at J_3_ for the *p*-channel mode and J_6_ for the *n*-channel mode increase exponentially with increasing temperature, as shown in Figure 5b. Moreover, at high temperatures, the recombination rates increase at J_2_ and J_4_ in the *p*-channel mode (at J_5_ and J_7_ for the *n*-channel mode) because the electrons and holes have sufficient kinetic energy to surpass the potential barrier, so that they recombine at the drain and source regions. This also accounts for the extensions of the drain and source described above.

To investigate the effect of temperature on the distribution of carriers in TG FBFETs, we analyzed the electron and hole concentrations in the intrinsic channel in the off state. Here, the electron and hole concentrations in the off state represent the thermally generated charge carriers. Charge carrier accumulation due to the high thermal energy in the *p*-channel mode was confirmed by the electron and hole concentrations shown in Figure 6a,b. The electron concentration in the potential well under the CG increases rapidly from 1.45 × 10^−8^ cm^−3^ to 1.12 × 10^19^ cm^−3^ as the temperature increases from −200 °C to 280 °C. Similarly, the hole concentration increases from 1.35 × 10^3^ cm^−3^ to 7.34 × 10^18^ cm^−3^ in the potential well under the PG close to the source region. Moreover, in the *n*-channel mode, a potential well for holes is formed in the CG and a potential well for electrons is formed in the PG close to the drain region, because the polarity of *V*_PG_ is opposite to that of the *p*-channel mode. As shown in Figure 6c,d, the electron concentration increases from 0.37 × 10^2^ cm^−3^ to 6.91×10^18^ cm^−3^, while the hole concentration increases from 0.47 × 10^2^ cm^−3^ to 1.01 × 10^19^ cm^−3^, with temperature increases from −200 °C to 240 °C. When the temperature is 280 °C, the high concentrations of electrons and holes throughout the channel indicate that the channel becomes conductive due to the collapse of the potential barrier caused by the thermally induced PF loop. Figure 6b,c show that the holes (electrons) from the drain (source) region injected via isotype junctions J_1_ (J_8_), as shown in Figure 4, contribute to the increase in the hole (electron) concentration in the *p*-channel (*n*-channel) mode in the channel region under the PG close to the drain (source) region.

## 4. Conclusions

In this study, we examined the effect of temperature on the operation mechanism of a TG FBFET through a TCAD simulation. The variations in the electrical characteristics of TG FBFETs over a wide temperature range from −200 °C to 280 °C are caused by the promotion and delay of PF loop occurrence. The promotion of the PF loop at high temperatures occurs via a thermal generation mechanism inside the channel. As the number of thermally generated carriers inside the channel increases, the potential barrier decreases; consequently, the latch-up voltage shifts negatively (positively) in the *n*-channel (*p*-channel) mode. By contrast, the decrease in the carrier concentration in the channel at low temperatures is responsible for the delay in the occurrence of the PF loop. The TG FBFET exhibits steep-switching characteristics with *SS* values of less than 6.6 mV/dec for the *n*-channel mode (7.2 mV/dec for the *p*-channel mode), regardless of the temperature. Moreover, the TG FBFET operates normally even in the range of −200~240 °C without failure, indicating its high reliability.

## Figures and Tables

**Figure 1 nanomaterials-14-00493-f001:**
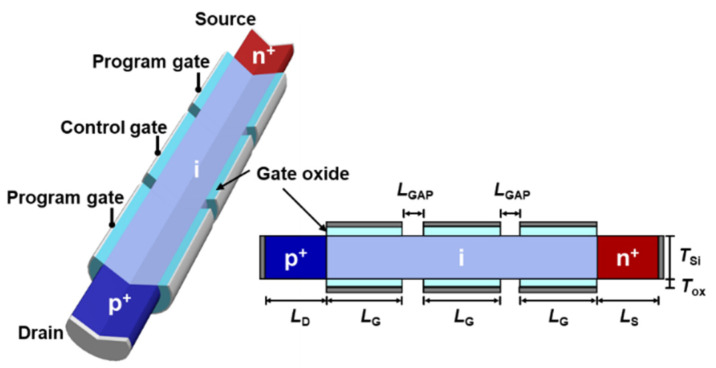
Schematic design and cross-sectional view of TG FBFET. The two-dimensional structure is used for temperature simulation.

**Figure 2 nanomaterials-14-00493-f002:**
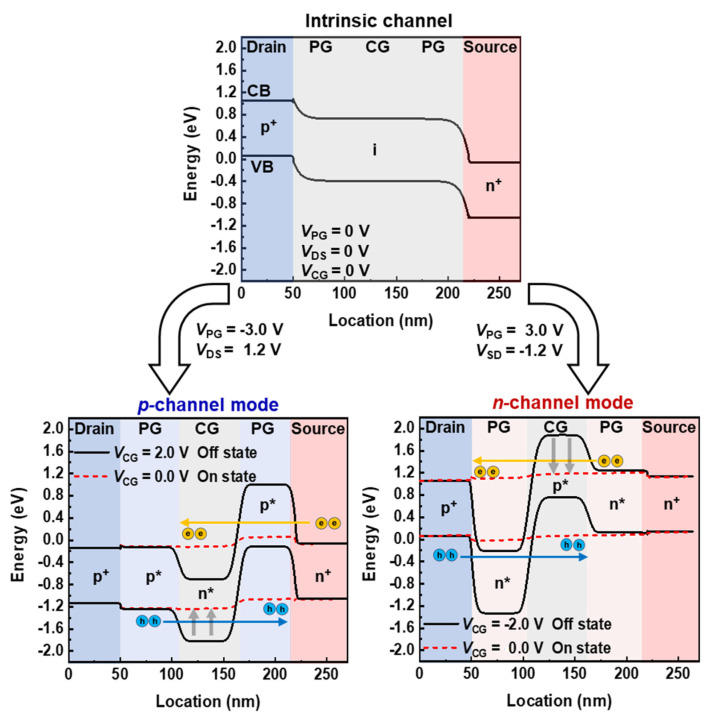
Energy band diagram of TG FBFET for intrinsic channel, *n*-channel, and *p*-channel modes.

**Figure 3 nanomaterials-14-00493-f003:**
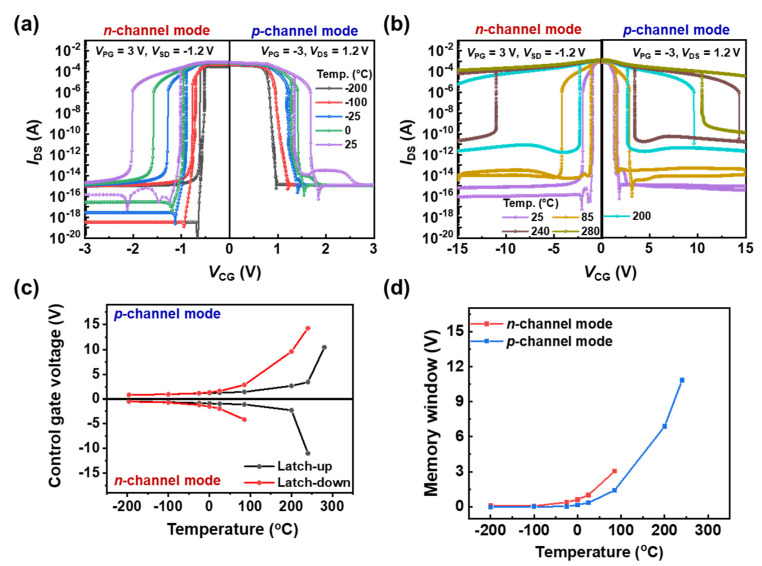
Transfer characteristics of TG FBFET in the *n*-channel and *p*-channel modes at (**a**) low and (**b**) high temperatures. Dependence of (**c**) latch-up/down voltages and (**d**) memory window on temperature change.

**Figure 4 nanomaterials-14-00493-f004:**
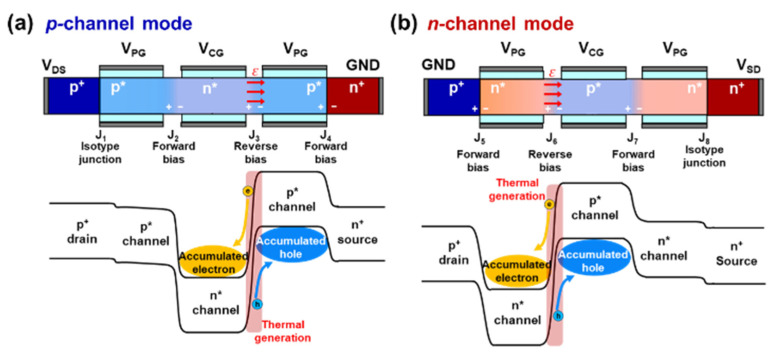
Mechanism of thermal generation and accumulation of electrons and holes in (**a**) *p*-channel and (**b**) *n*-channel mode operations.

**Figure 5 nanomaterials-14-00493-f005:**
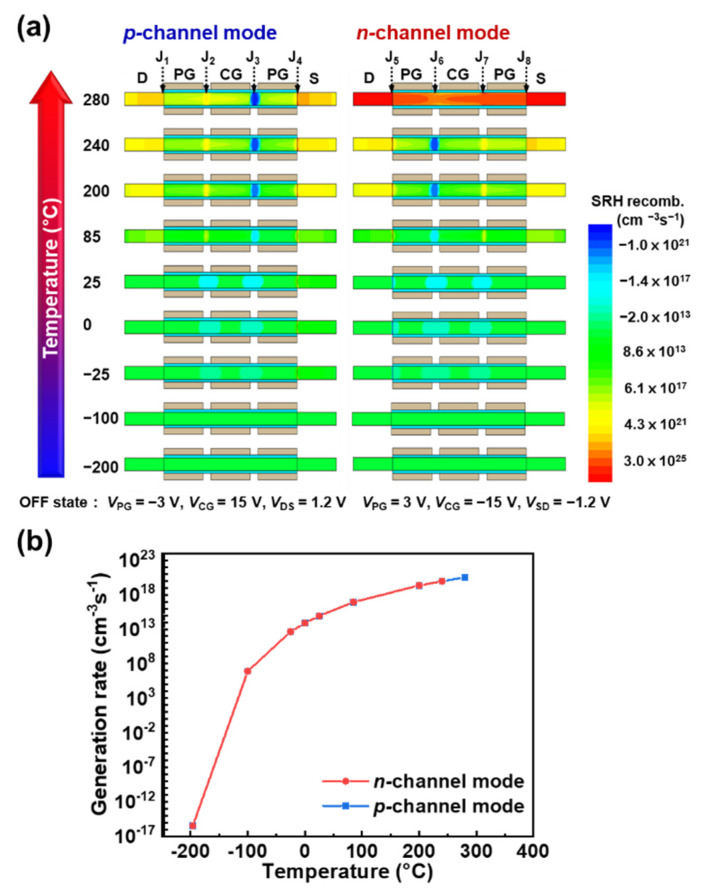
(**a**) Contour map of electron–hole generation in *p*-channel and *n*-channel modes at various temperatures. (**b**) Electron–hole generation rate at J_3_ for *p*-channel mode and J_6_ for *n*-channel mode.

**Figure 6 nanomaterials-14-00493-f006:**
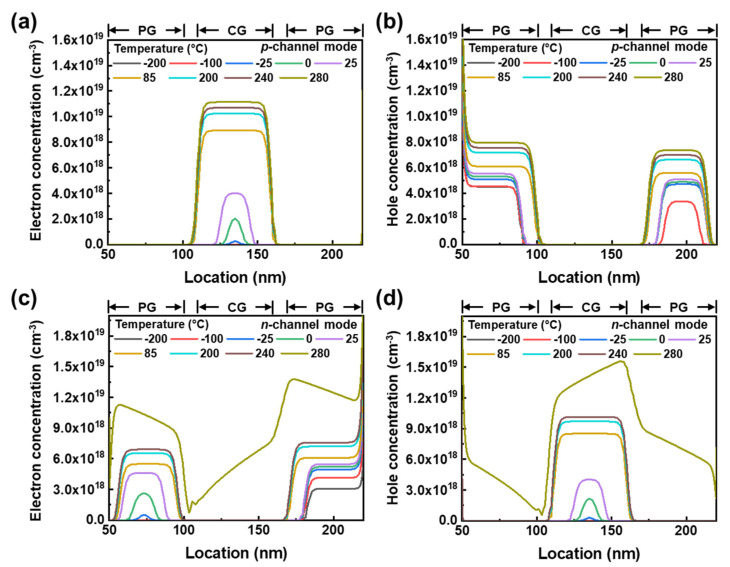
Carrier concentrations in channel region with increasing temperature. For *p*-channel mode, (**a**) electron concentration under CG, (**b**) hole concentration under PG. For *n*-channel mode, (**c**) electron concentration under PG, (**d**) hole concentration under CG.

## Data Availability

The data supporting the findings of this study are accessible upon request from the corresponding author.

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
