# Peer review of "Temperature-Dependent Feedback Operations of Triple-Gate Field-Effect Transistors"

_nanomaterials, 2024, doi:10.3390/nano14060493_

Round 1

Reviewer 1 Report

Comments and Suggestions for Authors

The authors report the simulated results for effect of temperature on the feedback operation of triple-gate feedback FET. The device shows a change in the latch voltage because of the interaction of thermally generated electron-hole and feedback loop occurrence, and results as well as in memory window. The device shows stability over a wide range of temperatures, which is potentially suitable for various temperature dependent applications. The reviewer has the following comments to be addressed.

Why was the oxide thickness kept so low about 2 nm? Usually, in experimental reports oxide thickness is kept around 300 nm. How do the simulation results vary with increasing oxide thickness?

Introduction lacks recent reports on temperature dependent investigation over p and n type 2D materials for memory window and mobility, and comparison with the simulated results in the discussion section. [cite: https://doi.org/10.1088/2632-959X/acbe11; https://doi.org/10.1016/j.mtnano.2023.100382].

It is possible to show the Cox depend on the gate voltage for the effect of charge trapping; as well as the mobility calculation in both the reverse and forward bias direction should be considered for the different Cox.

Reviewer 2 Report

Comments and Suggestions for Authors

The authors presented the effect of operation temperature on the TG-FBFET device mechanism through a TCAD simulation. The results are well-summarized and conclusive. A common for the authors before accepter this manuscript is can the authors adding some comments/descriptions that if a real TG-FBFET can show experimental data that match the simulation results the authors got here? 

Author Response

Our responses

We would like to thank you for reviewing our manuscript. Our detailed responses are as follows.

Reviewer #2 (Comments to the Author): The authors presented the effect of operation temperature on the TG-FBFET device mechanism through a TCAD simulation. The results are well-summarized and conclusive. A common for the authors before accepter this manuscript is can the authors adding some comments/descriptions that if a real TG-FBFET can show experimental data that match the simulation results the authors got here?

Our response:

Unfortunately, we cannot show experimental data for a real TG-FBFET because it is hard to fabricate a TG-FBFET with a nanowire structure in university laboratories. Nevertheless, we are confident that the simulation results reflect the experimental data since our previous study about the temperature effect and reliability analysis of a single-gate FBFET shows the same results for the simulation and the experiment [R1].

[R1] Park, T.; Lee, J.; Son, J.; Jeon, J.; Shin, Y.; Cho, K.; Kim, S. Temperature-Dependent Electrical Characteristics of p-Channel Mode Feedback Field-Effect Transistors. IEEE Access 2022, 10, 101458-101464.